# Microbial Contamination of Preservative-Free Artificial Tears Based on Instillation Techniques

**DOI:** 10.3390/pathogens11050592

**Published:** 2022-05-18

**Authors:** Jee-Hye Lee, Min-Ji Kang, Ha-Eun Sim, Je-Hyung Hwang

**Affiliations:** Department of Ophthalmology, Sanggye Paik Hospital, Inje University College of Medicine, Seoul 01757, Korea; jeehworld@naver.com (J.-H.L.); mjk4025@naver.com (M.-J.K.); s4451@paik.ac.kr (H.-E.S.)

**Keywords:** preservative-free artificial tears, contamination, eye drop instillation technique

## Abstract

Preservative-free artificial tears eliminate the side effects of preservatives but are prone to microbial contamination. This study evaluates the incidence of microbial contaminations in single-use vials of preservative-free 0.1% hyaluronate artificial tears. Based on what touched the vial tip during its first use, 60 unit-dose vials (0.5 mL) were divided into groups A (no touch, n = 20), B (fingertip, n = 20), and C (lid margin, n = 20). The vials were recapped after the first use, and the residual solution was cultured 24 h later. The solution from 20 aseptically opened and unused vials was also cultured (group D). Microbial contamination rates were compared between the groups using the Fisher’s exact test. Groups B and C contained 45% (9/20) and 10% (2/20) contaminations while groups A and D contained undetected microbial growth. The culture positivity rates were significantly different between groups A and B (*p* = 0.001) and groups B and C (*p* = 0.013) but not between groups A and C (*p* = 0.487). We demonstrate a significantly higher risk of contamination when fingertips touch the vial mouth. Therefore, users should avoid the vial tip touching the fingers or eyelid during instillation to prevent contamination of the eye drops.

## 1. Introduction

Artificial tears are a widely used lubricant for treating dry eye syndrome. Surfactant preservatives, such as benzalkonium chloride (BAK), added to these multidose eye drops help maintain their sterility after opening the cap. However, the preservatives can cause ocular irritation, punctate keratitis, toxicity to the corneal epithelial cells, and allergic reactions [1,2]. These side effects must be considered in patients with poor ocular surface integrity, as in dry eye syndrome. Therefore, preservative-free artificial tears have been introduced as an alternative to those with preservatives.

Preservative-free artificial tears are packed in unit-dose containers and recommended for a single use. Several studies have reported microbial contamination of preservative-free artificial tears with extended use in multidose [3,4] and unit-dose [5] containers. If the lid, conjunctiva, globe, or fingertip touch the vial mouth during instillation, it can lead to contamination [6,7,8]. This study compared the contamination rates in reclosable unit-dose vials of preservative-free artificial tears based on the instillation methods.

## 2. Results

Eighty single-use vials were evaluated, with 20 vials each in groups A, B, C, and D. In group B, 9/20 (45%) vials contained bacterial contamination, and the causative organisms were *Staphylococcus aureus* (n = 6, 30%) and *Staphylococcus epidermidis* (n = 3, 15%). In group C, 2/20 (10%) vials were contaminated with *Staphylococcus capitis* (n = 1, 5%) and *Staphylococcus caprae* (n = 1, 5%). No microbial contaminations were seen in groups A and D (Table 1).

Significant differences were seen in the culture positivity rates between groups A and B (*p* = 0.001) and groups B and C (*p* = 0.013) but not between groups A and C (*p* = 0.487) (Figure 1).

## 3. Discussion

Ophthalmic solutions are commonly contaminated with repeated use, and the administration of contaminated eye drops can cause severe ocular infections. Infectious keratitis, conjunctivitis, and even endophthalmitis have been reported due to bacterial growth [9,10,11,12,13]. Preservative-free eye drops are prone to microbial contamination once the vial is opened. Preservative-free, nonantibacterial eye drops are at an even higher risk of microbial contamination [5,14].

A few studies have examined the contamination of preservative-free artificial tears [5]. There is a risk of contamination at any point during the eye drop instillation [6,7,8] caused by the lid, conjunctiva, globe, or fingertip touching the vial’s mouth. In this study, we compared the incidence of microbial contamination in the residual eye drop solution depending on what touched the mouth of the unit-dose vial. Bacterial contamination was seen in 11/60 (18%) vials tested. While 9/20 (45%) vials in group B (fingertip touch) were contaminated with *Staphylococcus aureus* and *Staphylococcus epidermidis* growth, 2/20 (10%) vials in group C (eyelid touch) were contaminated with *Staphylococcus capitis* and *Staphylococcus caprae*. Kim et al. reported a 3.9% contamination rate using preservative-free hyaluronic acid 0.1% [5], which was lower than that of the current study. This is probably due to the shorter duration between first opening the vial and culturing the residual solution (10 h).

Group B had a significantly higher contamination rate than groups A and C, demonstrating that the fingertip touching the mouth of the vial is a major cause of contamination. Our findings are consistent with those of Kim et al., who reported that advanced age and fingertip touch are statistically significant risk factors for microbial contamination based on multivariate analysis [5].

Of the four identified microorganisms that caused contaminations in our study, three were Gram-positive, coagulase-negative *Staphylococcus* (CNS). CNS is a resident flora in the conjunctiva, eyelid, and skin, suggesting that eye drops can become contaminated when they contact the user’s hands, eyelid, or conjunctiva.

Preservative-free artificial tears in unit-dose vials are usually recommended to be used once and then discarded. However, since the vial usually contains more solution than is required for one application, users tend to reuse it multiple times by recapping it. Preservative-free artificial tears are especially susceptible to microbial contamination when a finger or hand touches the vial tip. Therefore, users need to be educated on the precise administration technique. Avoiding contact between the vial tip and any part of the eyelid, cilia, and hand will help reduce contamination of eye drops. Moreover, instilling eye drops at 90 degrees instead of 45 degrees also helps reduce contamination [15].

The study has some limitations, the first being the small sample size of this study. Second, the state of hand hygiene of the patients was not considered. Since hand hygiene can affect the microbial contamination rate, further study on this aspect is warranted. Third, quantitative measurements of bacterial loads were not performed.

## 4. Methods

This study analyzed eighty single-use vials containing 0.5 mL of preservative-free hyaluronic acid 0.1% (Hyalein Mini 0.1%^®^, Santen, Osaka, Japan). Participants who visited the Sanggye Paik Hospital Eye Clinic from January 2020 to June 2020, not diagnosed with conjunctivokeratitis or blepharitis, were each given an unopened single-use vial of hyaluronic acid 0.1%. They were asked to instill the eye drops as they usually did. Based on what touched the vial tip during opening and instillation, the vials were divided into group A (no touch, n = 20), group B (fingertips, n = 20), and group C (lid margin, n = 20). Patients between 20 and 40 years old of age who personally use eyedrops were included in the study. When the tips of eyedrops touched the patients’ hands or eyelids while applying them, the participants were instructed by the observer to immediately refrain from using the eyedrops and were guided in closing the eyedrop caps. Subsequently, the vials were brought to the clinic in a sealed container under observer guidance. When the number of participants in each group reached 20, participant recruitment was stopped. The vials were recapped and kept in the clinic at room temperature for 24 h, and any remaining fluid in the vials was cultured. Uninstilled preservative-free hyaluronic acid 0.1% was also cultured immediately after opening a vial (group D, n = 20). Aseptic sample handling was performed in the biosafety cabinet (BSC). The sample was placed in a liquid enrichment medium (thioglycolate broth) and kept in a 35 °C; incubator. Bacterial growth was observed with the naked eye every day during incubation. The incubation period was 14 days. The liquid medium turned turbid during bacterial growth; the liquid medium, which was suspicious of bacterial growth, was subcultured in the plate medium to check the presence of bacterial growth. On the 10th day of incubation, the suspected negative thioglycolate broth was incubated for 48 to 72 h by performing blind subcultivation on the biplate medium (BAP/MacConkey) agar, chocolate agar, and brucella agar. When the bacteria was cultured, Gram staining and identification was performed. When the bacteria was not cultured for 14 days, it was reported as “no growth.”

The study was approved by the Institutional Review Board (IRB) of the Sanggye Paik Hospital (NON2022-002). Informed consent was waived because the study evaluated the eye drops and not the patients. The contamination rates in the different groups were compared using Fisher’s exact test, and the *p*-value < 0.05 was considered statistically significant. Statistical analysis was performed using PASW Statistics ver. 18 (SPSS Inc., Chicago, IL, USA).

## 5. Conclusions

A fingertip touching the mouth of the vial significantly increases the risk of contaminating the contents of the vial. The most frequently identified microorganisms were Gram-positive CNS, which comprises the normal flora of the skin. Therefore, to avoid microbial contamination of preservative-free artificial tears, users should ensure that the tip of the vial does not touch the fingers or the eyelid during instillation. Furthermore, patients should be instructed in hand hygiene to reduce the likelihood of contamination. Unit-dose vials are recommended to be used only once after they are opened.

## Figures and Tables

**Figure 1 pathogens-11-00592-f001:**
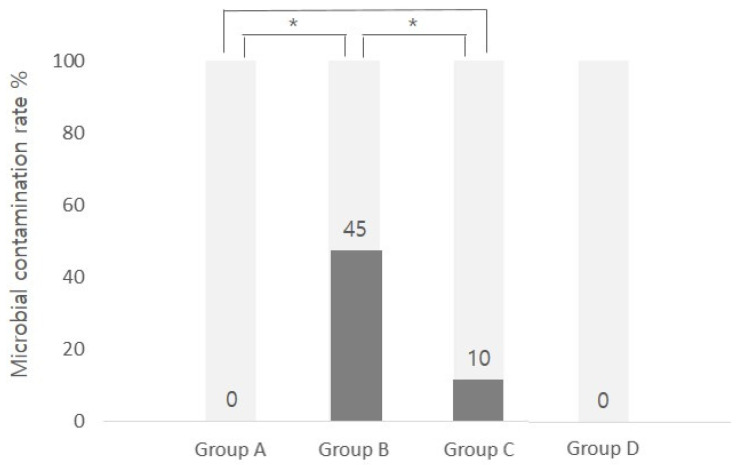
Contamination rates in the different groups. Microbial contamination rates were significantly different between groups A and B and groups B and C. However, no significant differences were seen between groups A and C (* *p*-value < 0.05). Group A: no-touch; Group B: fingertip touch; Group C: lid margin touch; Group D: uninstilled vial.

**Table 1 pathogens-11-00592-t001:** Microbial contamination of preservative-free artificial tears based on what touched the vial tip.

Group	n	Contaminated (n, %)	Identified Microorganisms (n)
A	20	0 (0)	
B	20	9 (45)	*Staphylococcus aureus* (6) *Staphylococcus epidermidis* (3)
C	20	2 (10)	*Staphylococcus capitis* (1) *Staphylococcus caprae* (1)
D	20	0 (0)	
Total	60 *	11 (18)	

Group A: no-touch; Group B: fingertip touch; Group C: lid margin touch; Group D: uninstilled vial. * Group D was excluded because the vials in this group were not instilled.

## Data Availability

The datasets used and/or analyzed during the current study are available from the corresponding author on reasonable request.

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
