# Peer review of "Microbial Contamination of Preservative-Free Artificial Tears Based on Instillation Techniques"

_pathogens, 2022, doi:10.3390/pathogens11050592_

Round 1

Reviewer 1 Report

Dear Authors,

Thank you for submitting your original work. This manuscript offers valuable information that supports the do's and don'ts of single-use preservative-free eye drops.

Although valuable, I notice that there is a lot of repetitive content in the introduction and discussion and that can be significantly improved by avoiding the repetition, while adding more relevant information from the existing literature.

In the methods section, please include some additional details like incubation media, temperature, and duration. Also, list out the steps involved in culture identification. Have there been any efforts to quantify the bacterial loads in the samples? If so, please add that information in the results section along with a graph.  Was the informed consent waiver approved by IRB? On what basis were the participants asked to instill the artificial tears if the informed consent was waived? How was the application of artificial tears justified to the participants?

While presenting the results have you considered including group D in the statistical analysis? If not, please include group D in the statistical analysis and kindly update the graph accordingly.

Specific comments to the authors are as below:

Line 39: Is it eight or eighty?

Line 68: Can you include the number of samples for lid margins or cilia touch separately in the methods section under group C?

Lines 73 to 77: Repetitive content

Line 86: Consider replacing 'infected' with contamination

Line 88: Avoid using 'we' in the sentence.

Line 89: What about the sites involved in contamination in the referred studies

Line 95 to 98: Even more relevant when you have the bacterial load information available.

Line 108: Very confusing as the authors state that this is a lab study even though the samples were collected in a hospital. Please clarify.

This manuscript needs to be revised as per the recommendations above to be able to accept for publication.

Wishing the authors all the best.

Author Response

I notice that there is a lot of repetitive content in the introduction and discussion and that can be significantly improved by avoiding the repetition, while adding more relevant information from the existing literature.

→ The repetitive content in the discussion has been deleted.

In the methods section, please include some additional details like incubation media, temperature, and duration. Also, list out the steps involved in culture identification. Have there been any efforts to quantify the bacterial loads in the samples? If so, please add that information in the results section along with a graph.  Was the informed consent waiver approved by IRB? On what basis were the participants asked to instill the artificial tears if the informed consent was waived? How was the application of artificial tears justified to the participants?

→ In the methods sections, additional information, such as the incubation media, temperature, duration, and culture identification steps, have been included. Quantitative measurements of the bacterial loads were not performed.

The patients who had been prescribed preservative-free 0.1% hyaluronate artificial tears were instructed to instill the artificial tears as they usually did.

While presenting the results have you considered including group D in the statistical analysis? If not, please include group D in the statistical analysis and kindly update the graph accordingly.

→ In our study, the contamination rates were compared based on what touched the vial tip,; thus, only groups A, B, and C (excluding group D) were statistically analyzed. Nevertheless, the Graph has been updated.

Specific comments to the authors are as below:

Line 39: Is it eight or eighty?

→ Eighty is correct.

Line 68: Can you include the number of samples for lid margins or cilia touch separately in the methods section under group C?

→ Samples of group C were not classified into lid margin touch and cilia touch since they cannot be recognized precisely. To avoid confusion, “lid margin or cilia touch” has been replaced with “lid margin touch.”

Lines 73 to 77: Repetitive content

→ The repetitive content has been deleted.

Line 86: Consider replacing 'infected' with contamination

→ 'infected' has been replaced with contamination

Line 88: Avoid using 'we' in the sentence.

→ The corresponding sentence has been revised accordingly.

Line 89: What about the sites involved in contamination in the referred studies

→ Cilia or globe, and fingertip were involved in contamination in Kim’s study.

Line 95 to 98: Even more relevant when you have the bacterial load information available.

→ Unfortunately, quantitative measurements of the bacterial loads were not performed. This has been mentioned in the limitations.

Line 108: Very confusing as the authors state that this is a lab study even though the samples were collected in a hospital. Please clarify.

→ The limitations have been revised.

Reviewer 2 Report

There some concerns about the paper that has not been evaluated and are key aspect of the study:

Introduction should be extended. First of all, there is no mention of the main question regarding contamination, that is hand hygiene. This is crucial and should be stated. Furthermore, it should be another group if the objective of the study is to teach users about the use of unidoses, as it is stated in the conclusions.

There isn´t neither any mention that apart from unidoses containers, there are other multidose preservative-free container as Aptar, Novelia, ABAK, etc.

Regarding methods, it should be stated how it was divided into groups as it is not possible to have the same amount in each group if the process is consecutive. Were the same patients used or were they different? How many patients participated? Patients washed their hand before instilling the drops?

If they weren´t consecutive patients, as it is supposed, results should state which is the percentage of each group.

Reference 15 is not correct. This is the original article which should be mentioned: https://www.ncbi.nlm.nih.gov/pmc/articles/PMC7414607/

Author Response

Introduction should be extended. First of all, there is no mention of the main question regarding contamination, that is hand hygiene. This is crucial and should be stated. Furthermore, it should be another group if the objective of the study is to teach users about the use of unidoses, as it is stated in the conclusions.

→ We were unable to consider hand hygiene in this study, since the patients did not especially washed their hands before instillation of eye drops. It could be either attributed to the structural limitation of the clinic, which was inconvenient for washing hands or their usual instillation manner. This point has been added to the limitations of this study.

There isn´t neither any mention that apart from unidoses containers, there are other multidose preservative-free container as Aptar, Novelia, ABAK, etc.

→ This study aimed to analyze the microbial contamination of unidose preservative-free artificial tears based on what touched the vial tip during opening and instillation. Multidose preservative-free artificial tears could be analyzed in future studies.

Regarding methods, it should be stated how it was divided into groups as it is not possible to have the same amount in each group if the process is consecutive. Were the same patients used or were they different? How many patients participated? Patients washed their hand before instilling the drops? If they weren´t consecutive patients, as it is supposed, results should state which is the percentage of each group.

→ Patients between 20 and 40 years old of age who personally use eyedrops were included in the study. When the tips of the eyedrops touched the paitents' hands or eyelids while applying them, the participants were instructed to immediately refrain from using the eyedrops, close the eyedrop caps and bring them to the clinic in a sealed container. When the number of the participants in each group reached 20, the participant recruitment was stopped.

We have added this information to the methods section.

Reference 15 is not correct. This is the original article which should be mentioned: https://www.ncbi.nlm.nih.gov/pmc/articles/PMC7414607/

→ Reference 15 has been revised.

Reviewer 3 Report

The manuscript refers to the study of the microbial contamination of eye drops. The manuscript is well organized but the applied methods were not described.

The following clarifications should be made:

Abstract

- Line 9 and 14: "Microbial" should added before "contamination".

- Line 16: "Contaminations" should be corrected to "it was not detected microbial growth".

Methods

- Line 39: "eight" correct to "eighty".

- The authors should detail the cultivation method applied. The culture media, the incubation temperature and period must be indicated. Also, the way the results will be expressed should be mentioned. 
The techniques applied for the identification of microorganisms must be indicated.

Results

Table 1 - the bacterial names should be italics. The meaning of asterisk should be indicated. Why total number is 60?

Discussion

Line 86 - the "S" of  "staphylococcus capitis" should be in capital letter.

Conclusion

The conclusion could be complemented with the information about the identified bacteria.

Author Response

Abstract

- Line 9 and 14: "Microbial" should added before "contamination".

→ The term "Microbial" has been added before "contamination".

- Line 16: "Contaminations" should be corrected to "it was not detected microbial growth".

→ "Contaminations" has been corrected to "it was not detected microbial growth".

Methods

- Line 39: "eight" correct to "eighty".

→ "eight" has been revised to "eighty".

- The authors should detail the cultivation method applied. The culture media, the incubation temperature and period must be indicated. Also, the way the results will be expressed should be mentioned.

The techniques applied for the identification of microorganisms must be indicated.

→ In the methods section, additional information, such as incubation media, temperature, duration, and culture identification steps have been included.

Results

Table 1 - the bacterial names should be italics. The meaning of asterisk should be indicated. Why total number is 60?

→ Table 1 has been revised. The meaning of the asterisk has been indicated mentioning why the total number is 60. (Group D was excluded because they were not instilled.)

Discussion

Line 86 - the "S" of "staphylococcus capitis" should be in capital letter.

→ the "S" of "staphylococcus capitis" has been revised accordingly.

Conclusion

The conclusion could be complemented with the information about the identified bacteria.

→ Information about the identified bacteria has been added.

Round 2

Reviewer 2 Report

As stated from the added information to methods, there are some comments who should be clarified. It is stated: "When the tips of the eyedrops touched the paitents' hands or eyelids while applying them, the participants were instructed to immediately refrain from using the eyedrops, close the eyedrop caps and bring them to the clinic in a sealed container"

Does it means that there is anyone controlling if the tip touches the finger or the eyelid and it is the own patient?

Does the patients also close the vial and could touch, in any case while closing, the tip?

Does the patients keeps the vial and carry the vial to the clinic? Does it occurs in the same day? 

In any case, and although it is not considered in the study, conclusions should state that hand-hygiene should be instructed to patients to reduce the likelihood of contamination and that unit-doses are only for one use or during the same day.

Author Response

Dear reviewer 2,

Thank you for your detailed review of our manuscript, which helped considerably improve the manuscript. Our point-by-point responses to your comments have been prepared and are included below. The corresponding changes in the manuscript are indicated using red font

As stated from the added information to methods, there are some comments who should be clarified. It is stated: "When the tips of the eyedrops touched the paitents' hands or eyelids while applying them, the participants were instructed to immediately refrain from using the eyedrops, close the eyedrop caps and bring them to the clinic in a sealed container"

Does it means that there is anyone controlling if the tip touches the finger or the eyelid and it is the own patient?

Response: Herein, the ophthalmology resident observed the professors’ patients during instillation and evaluated what part touched the vial tip; if the vial tip touched a foreign surface, the observer instructed the patients not to use the eyedrops. We included this information in the text as follows (page 3, lines 109–111):

 “When the tips of eyedrops touched the patients' hands or eyelids while applying them, the participants were instructed by the observer to immediately refrain from using the eyedrops and were guided in closing the eyedrop caps.”

Does the patients also close the vial and could touch, in any case while closing, the tip?

Response: The patients closed the vial under the guidance of the observer so that unnecessary touch was avoided. We included this information in the text as follows (page 3, lines 110–111):

“...the participants were instructed by the observer to immediately refrain from using the eyedrops and were guided in closing the eyedrop caps.”

Does the patients keeps the vial and carry the vial to the clinic? Does it occurs in the same day? 

Response: The vial was sealed in containers provided to the patients immediately after instillation and kept in the clinic for 24 hours. We included this information in the text as follows (page 3, lines 111–115):

“Subsequently, the vials were brought to the clinic in a sealed container under observer guidance. When the number of participants in each group reached 20, participant recruitment was stopped. The vials were recapped and kept in the clinic at room temperature for 24 hours, and any remaining fluid in the vials was cultured.”

In any case, and although it is not considered in the study, conclusions should state that hand-hygiene should be instructed to patients to reduce the likelihood of contamination and that unit-doses are only for one use or during the same day.

Response: Thank you for this comment; we have added this information in the Conclusions section as follows (page 4, lines 138–140):

“Furthermore, patients should be instructed in hand-hygiene to reduce the likelihood of contamination. Unit-dose vials are recommended to only be used once after these are opened.”

Round 3

Reviewer 2 Report

Thanks for the improvement